# One Model for One Graph: A New Perspective for Pretraining with Cross-domain Graphs

## Abstract

Graph Neural Networks (GNNs) have emerged as a powerful tool to capture intricate network patterns, achieving successes across different domains. However, existing GNNs require careful domain-specific architecture designs and training from scratch on each dataset, leading to an expertise-intensive process with difficulty in generalizing across graphs from different domains. Therefore, it can be hard for practitioners to infer which GNN model can generalize well to graphs from their domains. To address this challenge, we propose a novel cross-domain pretraining framework, "one model for one graph," which overcomes the limitations of previous approaches that failed to use a single GNN to capture diverse graph patterns across domains with significant gaps. Specifically, we pretrain a bank of expert models, with each one corresponding to a specific dataset. When inferring to a new graph, gating functions choose a subset of experts to effectively integrate prior model knowledge while avoiding negative transfer. Extensive experiments consistently demonstrate the superiority of our proposed method on both link prediction and node classification tasks.

## 1 Introduction

As a ubiquitous data structure, graphs can represent a wide range of structural data across different domains, such as academia (Yang et al., 2016), e-commerce (Ying et al., 2018; Borisyuk et al., 2024; Fan et al., 2019; Tang et al., 2020), and molecule (Ying et al., 2021). Graph neural networks (GNNs) have exhibited great performance when learning and inferring on a single graph dataset. However, most GNNs fail to generalize across domains due to the feature heterogeneity problem, in which graphs from different sources often contain node features with varying semantic meanings and dimensions.

Recently, feature dimension heterogeneity can be solved via two steps: (i) transform node features into textual descriptions (ii) employ Large Language Models (LLMs) to encode them into the aligned textual representation space. Multiple graph models (Liu et al., 2023a; Huang et al., 2023; Chen et al., 2024b;a) are then developed with inductive inference capability across graphs. Nonetheless, a recent benchmark (Chen et al., 2024b) reveals that, even within the aligned textual representation space, the positive transfer can only be found within the single domain, while the semantic disparity happens across different domains. Moreover, graphs from various domains exhibit significantly different structural properties. For example, the homophily property, a crucial factor affecting the node classification performance of GNNs, varies significantly across graphs. As noted by Mao et al. (2023), a single GNN struggles to capture varying levels of homophily simultaneously.

The aforementioned observations suggest that pretraining a single model for graphs from multiple domains is suboptimal. Therefore, in this work, we propose to individually pretrain one expert model for each pretraining graph and then save the set of different expert models as a model bank to effectively leverage cross-domain graphs. During inference, a subset will be automatically selected to produce a pre-trained model specfic to a test graph. This proposed pipeline is different from that adopted by the majority of existing cross-graph pretraining methods (Liu et al., 2023a; Huang et al., 2023; Chen et al., 2024b;a) as shown in Figure 1. The existing pipeline pretrains one model for all graphs and then applys the model to all test graphs as shown in Figure 1a which is referred to as the "one model for all graphs" pipeline. On the other hand, the proposed pipeline pretrains a bank of models with one model for each graph and automatically generates one model specific to a test graph

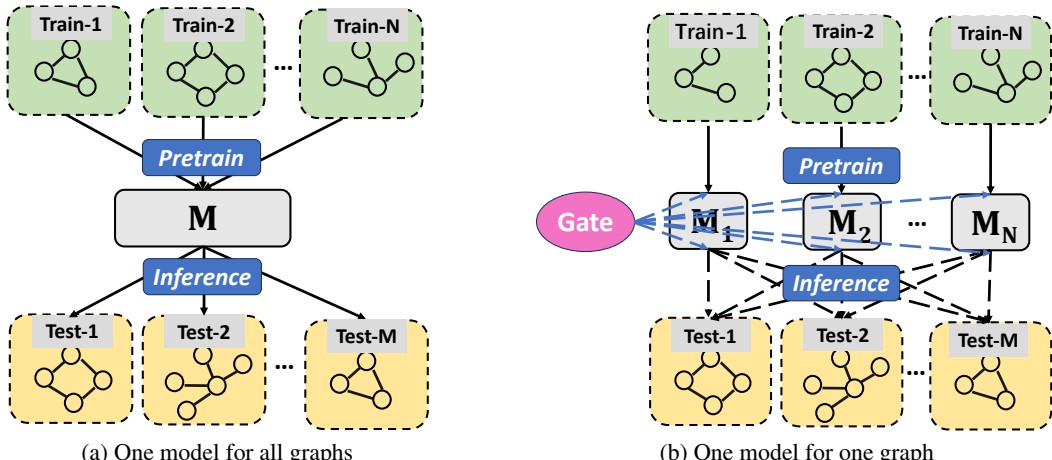

(a) One model for all graphs      (b) One model for one graph

Figure 1: Existing "one model for all graphs" pipeline vs. the proposed "one model for one graph" pipeline .

as shown in Figure 1b which is named as the "one model for one graph" pipeline. Compared to the "one model for all graphs" pipeline, the advantages of the "one model for one graph" pipeline are multifold. First, the new pipeline pretrains one model specific to one graph, which inherently reduces the feature and structural heterogeneity problems in cross-graph learning. Second, the new pipeline will produce one pretrained model specific to a test graph which potentially mitigates the negative transfer problem in the inference stage. Third, since each model is pretrained separately, the new pipeline makes it easier to incorporate new pretraining graphs, without needing to repeat the entire pretraining process. To enjoy the benefits of the new pipeline, we implement a novel "one model for one graph" pretraining framework with cross-domain graphs, OMOG. In OMOG, each expert model consists of a set of non-parametric SGCs (Wu et al., 2019) to capture information from different hops and an attention mechanism to fuse the information. After pre-training the expert, it fixes its parameters and trains a post-hoc gate on the same graph which is deployed to score the input data by its likeness to the source data with which the gate is trained. Thus, given a set of $N$ pretraining graphs, OMOG will train an expert and an associated gate on each graph, resulting in a bank of $N$ pretrained experts and $N$ gates. During the inference stage, every gate will give an relevance score for a test graph, and experts with top-k high corresponding scores will be fused to form a pre-trained model. Extensive experiments are conducted to demonstrate the superiority of our design on both zero-shot and few-shot learning settings.

## 2 RELATED WORK

**Cross-graph machine learning.** The graph machine learning community has recently witnessed a growing trend to extend models designed for a single graph across different graphs (or datasets) (Mao et al., 2024). The key obstacle to cross-graph learning stems from feature and structural heterogeneity. Early endeavors typically address feature heterogeneity by neglecting the original features (Qiu et al., 2020) and adopting GNN-based self-supervised learning to extract transferrable structural patterns. However, such a strategy performs poorly on text-rich networks and suffers from negative transfer (Xu et al., 2023) due to the structure shift across different datasets. Zhao et al. (2024a) adopt dimensionality reduction to unify the feature dimension while features remain poorly aligned. To generate high-quality unified features across graphs, LLM and prompt engineering(Liu et al., 2023a) have been adopted to generate features in a unified text space (Chen et al., 2024b). Liu et al. (2023a) focus on the cross-data co-training where the downstream dataset aligns with the pre-training one and achieves inferior performance when transferring to novel datasets (Chen et al., 2024b). Chen et al. (2024a); Tang et al. (2023) focus on transferring trained models to new datasets while presenting inferior performance with inadequate supervision. Li et al. (2024); Huang et al. (2023) identify the importance of reformulating prediction into nearest neighbor retrieval for effective prediction in low supervision scenarios, while their approach which uses one dataset to fit all graphs will struggle

when training across graphs and possibly suffer negative transfer. Xia & Huang (2024); Hou et al. (2024) further introduce a mixture-of-expert architecture to remedy this issue, while their gating function training lacks graph-aware supervision and adopts a fixed number of experts, resulting in inferior performance. He & Hooi (2024) adopt a LLM-based backbone, which incurs significant computational overhead. Zhao et al. (2024c) achieve cross-graph learning based on label space instead of feature space, which is orthogonal to our work.

Another line of work studies cross-task learning across graphs, where Jin et al. (2021); Ju et al. (2023) focuses on selecting pre-training tasks to adapt different downstream tasks, while Liu et al. (2023b); Sun et al. (2023; 2022) tackles the task heterogeneity to support different tasks with a unified backbone. Our work can potentially be combined with these works to support cross-task learning across graphs.

**Mixture-of-experts (MoE) on graphs.** Mixture-of-experts (Shazeer et al., 2017) has recently been adopted to graph machine learning to improve inference efficiency (Wang et al., 2024), enhance fairness (Liu et al., 2023c), tackle heterophily (Han et al., 2024; Zeng et al., 2023), and improve prediction performance by capturing diverse structural patterns (Hu et al., 2021; Ma et al., 2024). Our work extends the scope of MoE to multiple graphs. Combined with a suitable gate design, we achieve efficient and effective cross-graph learning, surpassing the performance of counterparts (Xia & Huang, 2024; Hou et al., 2024).

## 3 METHOD

In this section, we introduce our one model for one graph pretraining framework with cross-domain graphs, **OMOG**. It is an implementation of the new pipeline shown in Figure 1b. It consists of two stages – the pretraining stage and the inference stage. In the pretraining stage, OMOG will pre-train one model for one graph with one associated gate separately. In the inference stage, it will adaptively choose suitable experts for a test graph according to the associated gates and the test task. Before we detail these two stages, we start with introducing the problem formulation.

**Problem formulation**: In this work, we focus on text-attributed graphs (TAGs), or more generally, text-space datasets (Chen et al., 2024b) whose features can be converted into text-attributes. An input graph can be defined as $\mathcal{G} = (\mathcal{V}, \mathbf{A}, \mathbf{S})$, where $\mathcal{V} = \{v_1, v_2, ..., v_n\}$ is the set of $n$ nodes, and $\mathbf{A} \in \mathbb{R}^{n \times n}$ represents the adjacency matrix of the graph, and $\mathbf{S} = \{s_1, s_2, ...\}$ is the set of text descriptions for all nodes. We focus on cross-graph pretraining with a transferring setting. Specifically, assuming that we are given $N$ pretraining graphs $\{G_1, \cdots, G_N\}$, we would like to pretrain a model bank $\mathcal{M}$ with one model for each pretraining graph and then transfer knowledge in $\mathcal{M}$ to unseen test graphs. We focus on two downstream tasks: i.e., node classification and link prediction. For node classification, we aim to predict the category $y_i$ of the target node $v_i$. For the link prediction, we predict whether there is a link between two target nodes $v_i$ and $v_j$.

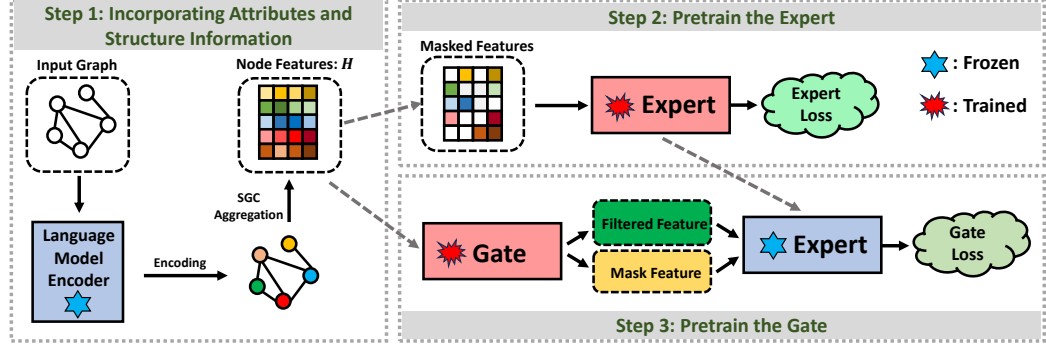

Figure 2: An illustration of the pretraining stage. The first step encodes the node attributes with language models and then applies SGC to incorporate the structure information. The second step pretrains the expert with contrastive loss. The third step trains a gate module to filter the domain-related features.

## 3.1 THE PRETRAINING STAGE

The whole pre-training process is illustrated in Figure 2, which contains the following steps:

1. **Incorporating attribute and structure information**: To achieve cross-graph pre-training across diverse domains, we first adopt LLMs to generate node features for each graph in a unified text space. Based on the unified feature space, we adopt non-parametric message passing (Wu et al., 2019) to generate node-level embeddings incorporating structural information.
2. **Pretraining graph-specific experts**: This step involves pre-training models that can effectively transfer to downstream datasets. As shown in (Xu et al., 2023), pre-training a single model across graphs with diverse structural properties results in negative transfer and catastrophic forgetting. Therefore, we design a model bank to preserve pre-training knowledge. This is achieved by pre-training one separate model for each graph.
3. **Pretraining gate modules**: To adaptively extract the proper experts for a test graph, we pre-train gate modules to determine the relevance between pre-trained models in the bank and test graphs. The pre-trained gate modules can then be applied to select the most relevant experts to produce a pretrained model specific to test graphs in the inference stage.

Next we introduce the technical details of these steps in the pretraining stage.

**Incorporating attribute and structure information.** A unified feature space is the requisite of cross-graph pre-training. As a result, we follow (Liu et al., 2023a; Chen et al., 2024b) to adopt LLMs as encoders to generate text embedding $\boldsymbol{x}_i$ based on node attributes $\boldsymbol{s}_i$. In this way, we get the node features $\mathbf{X} = \{\boldsymbol{x}_1, ..., \boldsymbol{x}_n\}$. Specifically, we adopt SentenceBERT (Reimers & Gurevych, 2019) which exhibits promising performance in previous studies (Chen et al., 2023; Liu et al., 2023a; Li et al., 2024).

Based on the unified feature space, we subsequently apply SGC (Wu et al., 2019) to integrate the graph structural information. First, we calculate the normalized adjacency matrix of the pretraining graph,

$$\mathbf{J} = \tilde{\mathbf{D}}^{-\frac{1}{2}} \tilde{\mathbf{A}} \tilde{\mathbf{D}}^{-\frac{1}{2}} \tag{1}$$

where $\tilde{\mathbf{A}} = \mathbf{A} + \mathbf{I}$ is the adjacency matrix with self-loop, and $\tilde{\mathbf{D}}$ is the degree matrix of $\tilde{\mathbf{A}}$. Then we could use $\mathbf{J}$ to update the node features with neighborhood information,

$$\mathbf{H}^{(\alpha)} = \mathbf{J}\mathbf{H}^{(\alpha-1)} \tag{2}$$

where we set $\mathbf{H}^{(0)} = \mathbf{X}$, and $\alpha$ is the number of neighborhood hops that are used to update the node features. By repeating the Equation 2, we can get node features $\mathbf{H}^{(\alpha)}$ integrated with different hops of structural information.

**Pretraining graph-specific experts.** As shown in (Xu et al., 2023), pre-training a single model across graphs with different structural properties leads to negative transfer, primarily due to the conflicts across graphs from diverse domains. To remedy this issue, we adopt a model bank to separately pre-train each model on each graph, which stores the pre-trained knowledge in each model. Considering the heterogeneous label space across different graphs, we construct a self-supervised pretext task to pretrain expert models with the learning objective adopted from Zhu et al. (2020).

Specifically, we first augment the node-level feature $\boldsymbol{h}_i = [\boldsymbol{h}_i^{(0)}, \boldsymbol{h}_i^{(1)}, ..., \boldsymbol{h}_i^{(\alpha)}]$ by randomly masking half of the features randomly following the method in Zhu et al. (2021), which results in two masked views $\hat{\boldsymbol{h}}_{i,0}$ and $\hat{\boldsymbol{h}}_{i,1}$. $\boldsymbol{h}_i^{(\alpha)}$ corresponds to the $i$-th row of the embedding matrix $\mathbf{H}^{(\alpha)}$. For $t$ nodes within a batch, views augmented from the same nodes are considered positive pairs, and those augmented from different nodes are considered negative pairs. To better capture the featurewise interaction, we adopt a transformer block (Vaswani, 2017) as the expert model backbone. The forward propagation process of expert can thus be represented as $\hat{\boldsymbol{f}}_{i,j} = \text{Expert}(\hat{\boldsymbol{h}}_{i,j})$, where the full definition of Expert is deferred to Appendix C. Expert is then optimized with the following contrastive loss:

$$\mathcal{L}_{expert} = \sum_{i=1}^{t} \log \frac{2e^{\text{sim}(\hat{\boldsymbol{f}}_{i,0}, \hat{\boldsymbol{f}}_{i,1})}}{\sum_{m=1}^{t}\sum_{n=1}^{t} e^{\text{sim}(\hat{\boldsymbol{f}}_{m,0}, \hat{\boldsymbol{f}}_{n,0})} + e^{\text{sim}(\hat{\boldsymbol{f}}_{m,1}, \hat{\boldsymbol{f}}_{n,1})} + 2e^{\text{sim}(\hat{\boldsymbol{f}}_{m,0}, \hat{\boldsymbol{f}}_{n,1})}} \tag{3}$$

Figure 3: An illustration of the inferene stage. We input the test graph features into each gate to calculate the relevance values to the corresponding experts. Them we select expert models with top-k largest values and fuse them into a new model to infer on the downstream tasks.

where $\text{sim}(\cdot, \cdot)$ is the operation to calculate the cosine similarity between two vectors.

**Pretraining the gate modules.** After keeping pre-trained knowledge in a model bank, we design a gating module to decide the relevance between the corresponding expert in the bank and downstream datasets during the inference stage. Specifically, the gate module aims to filter key graph features related to a domain. Relevant features after going through the gate should be near the domain's embedding cluster centroid, while unrelated features should be distant. To achieve this goal, we need an encoder to project the sample embeddings, a filter to refine the domain-related features, and a generator to produce negative samples. In our scenario, the well-trained expert can be reused as the encoder, thus we follow the idea in (Guo et al., 2023) and design a post-hoc gate. In OMOG, we employ an MLP for both the roles of filter and generator, which will learn to generate a matrix to mask the input features and thus leave out the domain-related patterns. Meanwhile, the mask matrix itself could be regarded as a negative sample since it is supposed to not have domain information. We train the MLP gate also in a mini-batch manner. When it takes in a feature $h_i$, it will generate an mask matrix by $a_i = \text{MLP}(h_i)$. Then the filtered feature is calculated as $\tilde{h}_i = h_i + a_i$, which is viewed as a positive sample of the domain. Meanwhile, the mask matrix $a_i$ is viewed as a negative sample. Then we will use the expert to encode $\tilde{h}_i$ and $a_i$ repectively, it will result in a positive embedding $\tilde{f}_i = \text{Expert}(\tilde{h}_i)$ and a negative embedding $o_i = \text{Expert}(a_i)$. We want the positive embedding to be close to the centroid of the domain embedding cluster while the negative embedding to be distant from the centroid $f_{center}$, that the training loss of the gate is designed as below,

$$\mathcal{L}_{gate} = \text{dis}(\tilde{f}_i, f_{center}) + \frac{1}{\text{dis}(o_i, f_{center})} \tag{4}$$

where $\text{dis}(\cdot, \cdot)$ is the Euclidian distance between two vectors, and $f_{center}$ can be calculated as $f_{center} = \text{MEAN}(\text{Expert}(\mathbf{H}))$.

## 3.2 THE INFERENCE STAGE

After pretraining a bank of experts and gates, we could adopt them to infer the unseen test data as shown in Figure 3. Similar to the forward propagation process of pretraining the gate, the feature will first be filtered by the gate and then encoded by the expert. Finally, the cosine similarity between the output and the centroid embedding of the domain will be calculated as a relevance score to indicate how likely the sample is related to the domain. For a test graph $\mathcal{G}_{test}$, we first get the node embeddings $\mathbf{H}_{test} = [\mathbf{H}_{test}^{(0)}, ..., \mathbf{H}_{test}^{(\alpha)}]$ aggregated by SGC. Subsequently, we will feed $\mathbf{H}_{test}$ into every gate to compute its domain-related representations. For the $p^{\text{th}}$ expert and gate which is trained on graph $\mathcal{G}_p$ with node embeddings as $\mathbf{H}_p$, the relevance score is calculated as follows:

$$v_p = \text{sim}(\text{MEAN}(\text{Expert}(\text{Gate}(h_{test}))), f_{center,p}) \tag{5}$$

where the $\text{sim}(\cdot, \cdot)$ is the operation to calculate the cosine similarity between two vectors, and $f_{center,p}$ can be calculated as $f_{center,p} = \text{MEAN}(\text{Expert}(\mathbf{H}_p))$.

After getting the relevance values for all experts, we will select top-k values with $\mathcal{E} = \text{top-k}(v_1, v_2, ...)$. Then we scale the weights with $\text{softmax}(\mathcal{E})$. Next we would use them to weight their corresponding expert models to produce a pretrained model.

Once the pretrained model is ready, we can use it to infer the target node feature $h_{test}$ and generate the output embeddings $f_{test}$. For zero-shot node classification, the label whose embedding has the highest cosine similarity with the test node output embedding is regarded as the prediction. For zero-shot link prediction, the logit of link existence is the cosine similarity between the two test nodes' output embeddings.

**Extension to few-shot learning setting.** For the few-shot learning, we follow the same process as zero-shot learning to produce a pretrained model. The key difference from zero-shot node classification is that we use both the label embedding and the centroid embedding of each class to compute the final predictions. Specifically, suppose that there are $s$ classes in the support sets, we input all the samples in the support set and calculate the average of output embeddings for each class. Thus, for each class, there is a centroid embedding $f_{avg,i}$, where $0 \leq i \leq s$. Suppose that the label embedding of each class is $l_i$, then the predicted label $y_{test}$ for a test node with output embedding $f_{test}$ can be calculated as following,

$$y_{test} = \text{argmax}_{0 \leq i \leq s}[\text{sim}(f_{test}, f_{avg,i}) + \text{sim}(f_{test}, l_i)] \tag{6}$$

where the $\text{sim}(\cdot, \cdot)$ is the operation to calculate the cosine similarity between two vectors.

## 4 EXPERIMENT

In this section, we conduct comprehensive experiments to evaluate the effectiveness of our proposed method OMOG from the following perspectives:

1. **RQ1:** Can our method effectively transfer pre-trained models to unseen test data in zero-shot and few-shot settings?
2. **RQ2:** How does each component of our method influence the transfer effectiveness?
3. **RQ3:** Why does expert gate selection notably enhance transfer effectiveness?

### 4.1 EXPERIMENTAL SETUP

**Datasets.** We utilize 10 diverse texture-attributed graphs sourced from Chen et al. (2024b). These datasets span a wide range of domains, including citation networks, social networks, and e-commerce networks. The graph sizes range from thousands to millions of nodes, with the number of classes across datasets spanning from 3 to 39. These datasets exhibit both domain shift and structural property shift (Chen et al., 2024b), effectively reflecting the challenges encountered when transferring pre-trained graph models to novel domains in real-world scenarios. For a comprehensive overview of the datasets, please refer to Appendix A.

**Evaluation settings.** To evaluate the effectiveness of our methods under the transferring setting (**RQ1**), we adopt a widely adopted setting that pre-trained models that are adapted to unseen test datasets with little (few-shot) or no downstream task supervision (zero-shot) (Chen et al., 2024b; Liu et al., 2023a). We consider both node classification and link prediction as target tasks. To test the transferring capability of models, we adopt a *leave-one-out* strategy. Specifically, given 10 adopted datasets, each time, one of them will be selected as the target downstream test data, and the other nine datasets will be used as the pre-training data. Regarding evaluation metrics, we adopt accuracy for node classification and Hits@100 for link prediction.

**Baselines.** To demonstrate the effectiveness of our framework, we consider state-of-the-art cross-graph learning baselines, which can be categorized as the "single model" and "mixture of models" frameworks.

- **"Single model"** framework adopts a unified backbone to achieve cross-graph learning, with representatives including OneForAll (Liu et al., 2023a), GCOPE (Zhao et al., 2024b), LLaGA (Chen et al., 2024a), ZeroG (Li et al., 2024) and Prodigy (Huang et al., 2023). Their major difference lies in the selection of backbone models, where OneForAll, ZeroG, GCOPE, and Prodigy are still based on GNNs, while LLaGA adopts LLM. Specifically, Prodigy and ZeroG transform the prediction into a nearest neighbor retrieval problem. Graph self-supervised learning baselines, including GCC (Qiu et al., 2020) and GraphMAE (Hou et al., 2022), also belong to this category.
- **"Mixture of models"** framework adopts a group of models to be pre-trained and then transferred to downstream tasks. Two representative methods including AnyGraph (Xia & Huang, 2024) and

GraphAlign (Hou et al., 2024). They directly apply the MoE architecture (Shazeer et al., 2017) without the correspondence between each model and each graph. Additionally, they utilize a fixed number of expert models, essentially functioning as a "multiple models for multiple graphs" approach.

The implementation details of our method can be found in Appedix D.

## 4.2 RQ1: Evaluating the transferability

In this subection, we evaluate the transferability of different cross-graph pretraining methods by comparing their performance on downstream tasks. Specifically, we focus on zero-shot and few-shot settings.

### 4.2.1 Transferring in a zero-shot setting

We first evaluate different cross-graph pretraining methods under zero-shot learning scenarios. We choose all baseline models applicable to the zero-shot learning settings, including OneForAll, LLaGA, AnyGraph, ZeroG, and our method. Since LLaGA adopts an LLM as the backbone model, it takes a considerably longer time to evaluate using the *leave-one-out* strategy. As a result, we pre-train it on Arxiv and Products and test it on every downstream task. To prevent data leakage when the target dataset is Product, we pre-train it using Arxiv and Sports. The results are shown in Table 1.

From Table 1, we make the following observations:

- **Our method consistently performs better in node classification and link prediction tasks.** Our method achieves the best performance on 8 out of 9 datasets for the node classification task and all of the datasets for the link prediction task, demonstrating that our method achieves superior transferability. In node classification and link prediction, our method outperforms the second-best baselines by a margin of 9%. Moreover, our method requires substantially less computation time than baselines like ZeroG, which requires fine-tuning the LLM.
- **Viewing zero-shot prediction as nearest neighbor retrieval is critical for effective zero-shot prediction.** Comparing the performance of each baseline, we find that the performance of OneForAll and LLaGA is consistently lower than other baselines. The key difference between these methods lies in their inference strategy. Specifically, OneForAll and LLaGA directly make inferences based on the model classification head, while other baselines project target embeddings and label embeddings into the same space for nearest neighbor retrieval.
- **Vanilla mixture-of-model can not solve the data heterogeneity problem effectively.** Despite adopting a mixture of model architecture, our model outperforms AnyGraph by a large margin, especially in node classification, which archives a 20% improvement on average. Comparing the design of these two models, our model presents two key distinctions: 1. we adopt one model for one graph; 2. we adopt an adaptive set of models when transferring to downstream tasks. These two designs make our models better tackle heterogeneity when transferring and achieve better performance. We detail the characteristics of our model mixture and gate design in Section 4.5.

Table 1: The transferring comparison under the zero-shot setting. Note that "NC" refers to node classification; "LP" refers to link prediction; and "Rank" is calculated based on the average rank of each model on each dataset. For results of more baselines, please refer to the appendix.

| Task | Methods | Child | History | Cora | Citeseer | Dblp | Products | Pubmed | Sports | Wikics | Rank |
|------|---------|-------|---------|------|----------|------|----------|--------|--------|--------|------|
| NC | Oneforall | 12.56 | 13.54 | 34.29 | 39.66 | 46.81 | 13.45 | 35.73 | 11.05 | 40.26 | 4.67 |
| | LLaGA | 13.75 | 14.58 | 33.78 | 40.79 | 47.53 | 17.26 | 35.38 | 12.35 | 39.37 | 4.22 |
| | AnyGraph | 13.84 | 15.16 | 55.63 | 40.03 | 50.27 | 22.36 | 37.92 | 15.35 | 50.84 | 3.00 |
| | ZeroG | 18.41 | 21.88 | 60.43 | 42.65 | 52.81 | 25.89 | **41.75** | 18.97 | 57.26 | 1.89 |
| | OMOG | **20.34** | **25.68** | **66.19** | **49.23** | **57.53** | **31.02** | 39.71 | **23.65** | **62.42** | **1.11** |
| LP | Oneforall | 15.28 | 10.83 | 17.46 | 16.52 | 13.31 | 13.77 | 15.35 | 14.30 | 15.83 | 4.78 |
| | LLaGA | 14.65 | 16.21 | 18.01 | 19.66 | 17.43 | 12.64 | 17.81 | 15.87 | 21.27 | 4.22 |
| | AnyGraph | 26.24 | 28.63 | 54.24 | 47.94 | 49.64 | 33.76 | 46.93 | 32.59 | 49.82 | 2.11 |
| | ZeroG | 21.83 | 24.39 | 49.36 | 43.18 | 41.08 | 31.27 | 40.28 | 33.98 | 45.19 | 2.89 |
| | OMOG | **31.29** | **34.86** | **56.28** | **50.72** | **53.46** | **40.95** | **49.42** | **37.81** | **52.38** | **1.00** |

### 4.2.2 Transferring in a few-shot setting

We then evaluate different cross-graph pretraining methods under the few-shot setting. We consider all applicable baselines, including GCC, GraphMAE, OneForAll, LLaGA, GraphAlign, GCOPE, and Prodigy. For OneForAll and Prodigy, we follow (Liu et al., 2023a) to augment the target prediction subgraph with graph prompts sampled from each class. GraphAlign and our method use the inference strategy introduced in Section 3.2 to generate label embeddings for each class. Other baseline methods directly adopt the few-shot labels as supervision to fine-tune the prediction head. For Prodigy, we follow the original setting to pretrain the model on a subset of the MAG240M dataset. Considering that most baseline methods are designed for node classification, we present the results for few-shot node classification. For each dataset, we randomly select 5 samples for each class.

As shown in Table 2, we summarize the main observations below:

- **Our method outperforms other baseline methods**. Our method performs best on 8 of the 10 downstream datasets. Comparing our method to the best baseline Prodigy, our method significantly outperforms it on heterophilous dataset, i.e., Ratings. Our method achieves more than 5% improvement compared to Prodigy despite using less pre-training data. This demonstrates our method's transferability to unseen downstream datasets with different structural properties.
- **Our methods demonstrate more superiority on datasets with complicated label space.** Only on Cora and Pubmed, whose class numbers are 3 and 5, respectively, the performance GraphAlign can slightly surpass our method. For more complicated cases where the class number of the dataset is more than 10, our method consistently outperforms other baselines. Compared to GraphAlign, which also adopts a "mixture of model" design, our method achieves an improvement of over 6% on average.

Table 2: The results for few-shot node classification. Note that "Rank" is calculated based on the average rank of each model on each dataset.

| Methods | Ratings | Child | History | Cora | Citeseer | Dblp | Products | Pubmed | Sports | Wikics | Rank |
|---|---|---|---|---|---|---|---|---|---|---|---|
| GCC | 23.25 | 17.86 | 18.14 | 33.28 | 35.62 | 34.52 | 21.04 | 35.11 | 16.48 | 29.93 | 7.7 |
| GraphMAE | 22.68 | 18.74 | 19.94 | 35.79 | 37.20 | 38.18 | 20.87 | 36.34 | 18.42 | 28.87 | 7.3 |
| Oneforall | 26.73 | 27.81 | 26.59 | 56.26 | 40.27 | 46.24 | 31.27 | 39.93 | 23.91 | 41.74 | 5.8 |
| LLaGA | 31.51 | 29.26 | 27.28 | 53.23 | 42.15 | 43.28 | 32.86 | 40.27 | 25.22 | 43.37 | 5.1 |
| GraphAlign | 34.79 | 32.69 | 32.71 | 72.86 | 52.39 | 58.60 | 44.62 | **50.76** | 32.65 | 63.17 | 3.4 |
| GCOPE | 37.85 | 32.73 | 36.29 | 72.17 | 55.87 | 60.24 | 46.02 | 48.10 | 35.88 | 59.28 | 2.8 |
| Prodigy | 30.88 | 33.63 | 35.82 | **77.59** | 56.28 | 60.83 | 45.35 | 44.87 | 33.18 | 64.23 | 1.9 |
| **OMOG** | **39.23** | **35.87** | **38.25** | 75.41 | **59.36** | **63.24** | **46.27** | 49.82 | **36.72** | **65.39** | **1.1** |

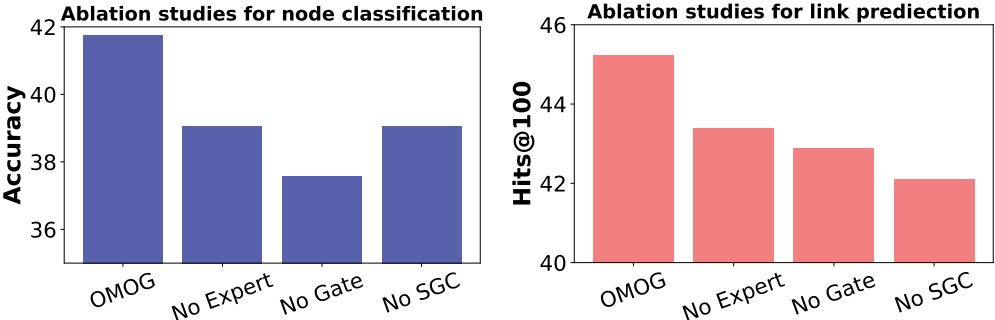

Figure 4: The impact for key components on OMOG

### 4.3 RQ2: Ablation Study

We then study how each key component of OMOG affects the transferring effectiveness to answer **RQ2**. We identify three key components of OMOG:

- **Expert module** acts as the backbone model to solve the prediction task for each graph.
- **SGC module** generates node-level graph embeddings using message passing.

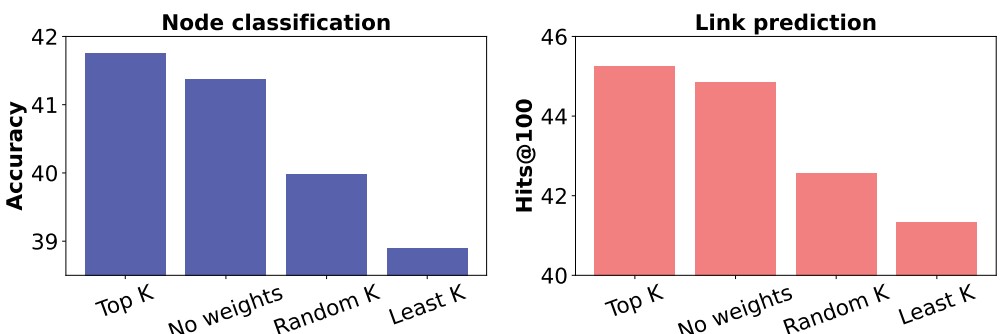

Figure 5: The performance of different gate designs.

- **Gate module** takes node-level embeddings as input and generates a relevance score to select the related experts.

As shown in Figure 4, we find that

1. **Every component contributes to effective transferring.** This ablation study reveals that each component significantly contributes to the model's overall performance.
2. **Gating mechanism is crucial to cross-graph node classification.** For node classification, we find that removing the gating mechanism results in a significant performance drop, which suggests that gating plays an important role in addressing data heterogeneity by adaptively selecting experts from the proper domain. As a comparison, SGC components play the most important role in link prediction, which means structural information is vital for link prediction.
3. **LLM embedding plays an important role.** When solely using the aggregated LLM embedding for prediction, the model can still have good performance, indicating the importance of aligned feature space in the zero-shot learning scenario.

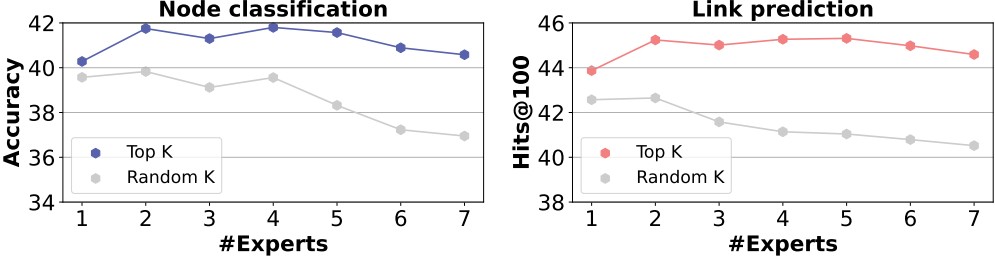

Figure 6: The effect of the number of experts.

### 4.4 RQ3: INVESTIGATING THE GATE DESIGN

Considering the importance of expert selection and distinguishing our work from existing ones (Hou et al., 2024; Xia & Huang, 2024), we further study the influence of different expert selection strategies to answer **RQ3**. We compare the following strategy variants to our original design stated in Section 3: (1) "No weights" still adopts the TopK selection strategy while removing the weights for each expert; (2) "Random K" randomly selects $K$ experts instead of experts with highest scores; and (3) "Least K" selects $K$ experts with lowest scores.

As shown in Figure 5, we observe that the original gate design in OMOG outperforms all variants, which suggests that

- The score learned by our gating module can guide us to select the most helpful experts for transferring, and consequently making the "TopK" strategy outperforms the "Random K" and "Least K" strategies.

- The weight given by the gating module can further help the fusing of selected experts, which makes the "TopK" strategy outperforms the "No weights" strategy.

Furthermore, we check how the number of selected experts $K$ affects the performance and the results are shown in Figure 6: with the increase of the number of experts, the performance of "Top K" first increases and then becomes relatively stable while the performance of "Random K" will consistently decrease, indicating significant negative transfer. This observation supports that the gate selection in our design can help mitigate the negative transfer when including more pretraining graphs.

### 4.5 CASE STUDY

To visually demonstrate how expert selection addresses the data heterogeneity problem in cross-graph pretraining, we present a case study that investigates how our proposed gating functions adaptively select proper experts based on downstream datasets. We consider the 9 downstream datasets for zero-shot node classification. Specifically, we the ego-subgraphs of 10 randomly sampled nodes from each dataset and visualize the average relevance score given by different gate functions. As shown in Figure 7, we observe that gates pre-trained on datasets similar to the target dataset exhibit higher scores. For instance, when Cora is the target, gates pre-trained on Citeseer and Dblp assign higher scores, likely because all three datasets are citation graphs within computer science.

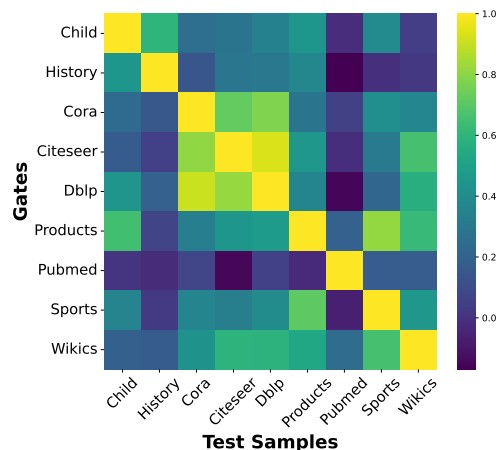

Figure 7: A case study on the gate selection.

## 5 CONCLUSION

In this paper, we present a new perspective together with an easy yet effective framework, "one model for one graph" (**OMOG**), to achieve effective cross-graph learning. Through extensive experiments, we develop the following practices for cross-graph learning: training one expert model for each graph and then utilizing pre-trained gate functions to select the experts most proper for downstream tasks adaptively. Our perspective can benefit future development in related areas, such as graph foundation models.

## 6 REPRODUCIBILITY STATEMENTS

To enhance the reproducibility of our methods, we detailed the implementations in Appendix D. The codebase and implementation details can also be found in the anonymous github link `https://anonymous.4open.science/r/duehfeuifoewhfowe/`.

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

## A  DATASET DETAILS

For the detailed statistics of the datasets we use in the experimets, we record them in Table 3.

Table 3: Details of our datasets.

| Name | #Graphs | #Nodes | #Edges | Domains | Tasks | #Classes | Metrics |
|---|---|---|---|---|---|---|---|
| **Cora** | 1 | 2708 | 10556 | CS Citation | Node, Link | 7 | Accuracy, Hits@100 |
| **CiteSeer** | 1 | 3186 | 8450 | CS Citation | Node, Link | 6 | Accuracy, Hits@100 |
| **Arxiv** | 1 | 169343 | 2315598 | CS Citation | Node, Link | 40 | Accuracy, Hits@100 |
| **History** | 1 | 41551 | 503180 | E-commerce | Node, Link | 12 | Accuracy, Hits@100 |
| **Child** | 1 | 76875 | 2325044 | E-commerce | Node, Link | 24 | Accuracy, Hits@100 |
| **Sportsfit** | 1 | 173055 | 3020134 | E-commerce | Node, Link | 13 | Accuracy, Hits@100 |
| **Products** | 1 | 316513 | 19337722 | E-commerce | Node, Link | 39 | Accuracy, Hits@100 |
| **Amazon Ratings** | 1 | 24492 | 186100 | E-commerce | Node, Link | 5 | Accuracy, Hits@100 |
| **Pubmed** | 1 | 19717 | 88648 | Bio Citation | Node, Link | 3 | Accuracy, Hits@100 |
| **WikiCS** | 1 | 11701 | 431726 | Knowledge | Node, Link | 10 | Accuracy, Hits@100 |
| **DBLP(*)** | 1 | 14376 | 431326 | CS Citation | Node, Link | 4 | Accuracy, Hits@100 |

## B  NUMERICAL RESULTS

We record the numerical values of the ablation study of each dataset in the Table 6

Table 4: The results for zero-shot node classification.

| Task | Methods | Child | History | Cora | Citeseer | Dblp | Products | Pubmed | Sports | Wikics |
|---|---|---|---|---|---|---|---|---|---|---|
| **NC** | **No SGC** | 17.23 | 23.91 | 61.73 | 47.82 | 52.74 | 29.56 | 37.62 | 20.47 | 60.35 |
| | **No Gate** | 16.03 | 23.29 | 60.84 | 45.11 | 53.98 | 27.17 | 34.76 | 18.12 | 58.72 |
| | **No Expert** | 18.83 | 23.42 | 64.43 | 46.93 | 53.89 | 28.63 | 36.22 | 19.47 | 59.73 |
| | **OMOG** | 20.34 | 25.68 | 66.19 | 49.23 | 57.53 | 31.02 | 39.71 | 23.65 | 62.42 |
| **LP** | **No SGC** | 28.63 | 31.30 | 54.18 | 49.57 | 49.76 | 37.43 | 45.84 | 34.92 | 47.26 |
| | **No Gate** | 29.11 | 32.61 | 54.84 | 48.63 | 48.40 | 38.57 | 46.62 | 35.86 | 51.28 |
| | **No Expert** | 29.75 | 33.92 | 55.28 | 49.71 | 50.08 | 37.93 | 47.38 | 35.94 | 50.48 |
| | **OMOG** | 31.29 | 34.86 | 56.28 | 50.72 | 53.46 | 40.95 | 49.42 | 37.81 | 52.38 |

## C  COMPUTATION

The computation formula for the expert model is shown as follow: Given input $H$ with shape $(B, \alpha, d)$, where $B$ represents the batch size, $\alpha$ represents the number of $\alpha$ SGC heads, and $d$ represents the hidden dimension. Then, $H_o = \text{softmax}\left(\frac{HW_q(W_k^T H^T)}{\sqrt{d_k}}\right) HW_v$, $H_1 = \text{LayerNorm}(H + H_o)$, and finally $\hat{f} = \text{LayerNorm}(H_1 + \text{MLP}(H_1))$.

## D  IMPLEMENTATIONS

In this section, we present our detailed implementations of OMOG. For the vector length of language embeddings, we set them to 384 to balance the efficiency and performance. For the number $\alpha$ of SGC operations, we set it to 4. And we choose top-2 models in the fusion stage to select 2 experts which largest relevance scores with the downstream task.

In the pretraining stage for experts and gates, we use Adam (Kingma, 2014). The initial learning rate is set to be 0.0001.

## E  RESULTS OF MORE BASELINES

We include the results of more baselines of zero-shot node classification and link prediction tasks.

Table 5: The results for zero-shot learning for more baselines.

| Task | Methods | Child | History | Cora | Citeseer | Dblp | Products | Pubmed | Sports | Wikics |
|------|---------|-------|---------|------|----------|------|----------|--------|--------|--------|
| NC | GraphMAE | 15.37 | 20.63 | 62.83 | 46.78 | 51.27 | 26.23 | 33.95 | 20.72 | 57.98 |
| | GCOPE | 16.83 | 20.37 | 61.48 | 44.26 | 53.40 | 27.75 | 34.26 | 19.88 | 55.23 |
| | GraphAlign | 18.63 | 26.39 | 63.45 | 45.11 | 55.11 | 30.82 | 37.43 | 21.73 | 60.17 |
| | OMOG | 20.34 | 25.68 | 66.19 | 49.23 | 57.53 | 31.02 | 39.71 | 23.65 | 62.42 |
| LP | GraphMAE | 22.35 | 24.15 | 51.68 | 43.78 | 52.28 | 33.21 | 42.15 | 34.20 | 49.24 |
| | GCOPE | 24.33 | 25.83 | 50.24 | 44.84 | 47.24 | 34.85 | 45.53 | 33.54 | 47.29 |
| | GraphAlign | 31.81 | 32.28 | 52.92 | 51.34 | 51.21 | 38.05 | 44.28 | 35.53 | 50.12 |
| | OMOG | 31.29 | 34.86 | 56.28 | 50.72 | 53.46 | 40.95 | 49.42 | 37.81 | 52.38 |

## F  RESULTS OF OTHER PRETRAINING METHOD

In the table below we attach the results attained by switching self-supervised approaches to Graph-MAE. As a comparison, we find that both the original GRACE method and GraphMAE methods could attain similar performance.

Table 6: The results for using GraphMAE to pretrain the experts.

| Methods | Child | History | Cora | Citeseer | Dblp | Products | Pubmed | Sports | Wikics |
|---------|-------|---------|------|----------|------|----------|--------|--------|--------|
| GraphMAE | 20.78 | 26.51 | 64.28 | 49.03 | 56.88 | 31.97 | 39.89 | 23.65 | 61.54 |
| Grace | 18.63 | 26.39 | 63.45 | 45.11 | 55.11 | 30.82 | 37.43 | 21.73 | 60.17 |

