# OpenReview forum: "One Model for One Graph: A New Perspective for Pretraining with Cross-domain Graphs"
_ICLR.cc/2025/Conference — Submitted to ICLR 2025_

### Official Review · Reviewer_zkPv · 2024-10-24

**Soundness:** 2
**Presentation:** 2
**Contribution:** 2
**Rating:** 5
**Confidence:** 3

**Summary:**

This paper proposes OMOG to pretrain one model for one graph in cross-domain transformation. OMOG uses SGC as message aggregation before experts learning, and uses the contrastive method for expert and gate training. In the inference stage, OMOG activates the top-k of experts to infer node labels. The experimental results proved the effectiveness of the method.

**Strengths:**

1. The approach of building an expert model for each graph intuitively addresses the issue of negative transfer in graph pre-training.
2. The problem addressed in this paper is a crucial part of foundational research on graph models and is currently a topic of significant interest among researchers.
3. The experiments involve graph data from multiple domains and include comparisons with recent, noteworthy methods in graph transfer learning.

**Weaknesses:**

1. According to the method statement, graphs from different domains are required to have input spaces of the same size, which seems difficult to satisfy with real-world data.
2. The construction of multiple experts for input graphs appears to be relatively naive, as it merely involves repeating several encoders and using similarity ranking with a central vector for averaging activation.

**Questions:**

1. What is the means of t in Equation (3)? How do you ensure that each expert has different parameters? Based on Equation (3), it seems that each expert has the same input and parameters, leading to identical outputs. What, then, is the purpose of having multiple experts?
2. In line 241, the mask matrix seems to be replaced by an MLP. Why are the node embeddings transformed by the MLP considered to be negative embeddings?
3. In Equation (4), is f_center ​the average output of a single graph across multiple experts, or the average output of multiple source graphs through their respective experts? How does this function as an anchor point for training the gate?
4. What are the parameters of the Gate in Equation (5)? Why is the Gate used before the Expert?
5. According to the inference process, an unseen graph activates the top k experts based on the highest correlation between each expert's output and the output average. How do you ensure that the pre-trained graph domain encompasses a sufficiently heterogeneous feature space to handle all potentially unseen graph domains?
6. How is the total number of experts determined, especially when the number of graph domains during pre-training and testing is uncertain?

---

> ### Author Response · Authors · 2024-11-20
>
> >**Q1** What is the means of t in Equation (3)? How do you ensure that each expert has different parameters? Based on Equation (3), it seems that each expert has the same input and parameters, leading to identical outputs. What, then, is the purpose of having multiple experts?
>
> **A1.** Thank you for your question. In Equation (3), $t$ indicates the number of training nodes in a batch. Equation (3) indicates the loss calculation to train a single expert for one graph. Each expert has the same architecture but is trained on different datasets. Thus, their inputs and outputs are different. In other words, for one graph, we pretrained one distinct model.
>
>
> >**Q2.** In line 241, the mask matrix seems to be replaced by an MLP. Why are the node embeddings transformed by the MLP considered to be negative embeddings?
>
> **A2.** Thank you for your question. For the MLP-generated mask matrix $a_i$, we tend to let it learn to mask the the domain-irrelevant features through the two loss terms $dis(\tilde{f},f_{center})$ and $\frac{1}{dis(\mathbf{o_i,f_{center}})}$ in Equation (4), where $\tilde{f}$ is embedding of original features masked by $a_i$ in the output space, $o_i$ is embedding of $a_i$ in the output space, and $f_{center}$ can be viewed as the representative feature of the training domain. With the loss terms decreasing, the distance between the mask matrix $o_i$ and $f_{center}$ will become larger but $\tilde{f}$ masked by $o_i$ will become closer to the $f_{center}$. Hence, $a_i$ are supposed to learn to mask the domain-irrelevant features of the input vector.
>
> >**Q3** In Equation (4), is f_center the average output of a single graph across multiple experts, or the average output of multiple source graphs through their respective experts? How does this function as an anchor point for training the gate?
>
> **A3** Thank you for your question. $f_{center}$ is the average out of one source graph through its own expert. It is a centroid point for the feature distribution of the source graph. Thus, we could view the feature vectors close to $f_{center}$ as in the similar domains of the source graph, while vectors distant from $f_{center}$ in different domains. In this way, $f_{center}$ serves as an anchor point for training the gate as we described in the answer for the last question.
>
>
> >**Q4.** What are the parameters of the Gate in Equation (5)? Why is the Gate used before the Expert?
>
> **A4.** Thank you for your question. The Gate and Expert in Equation (5) are pretrained on the $p$th source dataset following the steps in Section 3.1. The gate is used before the Expert because Equation (5) is used to calculate the relevance score with the help of graph.
>
> >**Q5.** According to the inference process, an unseen graph activates the top k experts based on the highest correlation between each expert's output and the output average. How do you ensure that the pre-trained graph domain encompasses a sufficiently heterogeneous feature space to handle all potentially unseen graph domains?
>
> **A5.** Thank you for your questions. We agree that there is possibility that the framework will have difficulty if encountering datasets from unseen domains. Actually, the out-of-distribution problem is a general problem for pretraining models, even for LLMs [1,2]. To deal with the data heterogeneity problem, researchers usually add data from more sources to pretraining datasets. In our case, we try to include the typical datasets from different graph domains to enlarge the scope of the model bank. In this way, the framework could find similar models for most of the common downstream tasks.
>
> Reference:
>
> [1] Unsupervised Out-of-Domain Detection via Pre-trained Transformers; Xu et al.
> [2] Multi-Level Knowledge Distillation for Out-of-Distribution Detection in Text; Wu et al.
>
> >**Q6.** How is the total number of experts determined, especially when the number of graph domains during pre-training and testing is uncertain?
>
> **A6.** Thank you for your question. For the pretraining datasets, we select representative datasets from common domains to construct the pretrained model bank. Hence, the number of experts in the model bank is the same as the number of pretraining datasets. For the number of experts in downstream tasks, we choose top-k relevant experts from the whole model bank. Specifically, we conduct experiments as shown in Figure 6, which reveals that 2 or 3 experts could be an ideal choice that boosts the useful knowledge to transfer and mitigate the negative transfer.

---

> > ### Comment · Reviewer_zkPv · 2024-11-27
> >
> > Thank you for your response, my concerns are well addressed. I will raise my score.

---

### Official Review · Reviewer_6z4L · 2024-11-02

**Soundness:** 2
**Presentation:** 2
**Contribution:** 2
**Rating:** 5
**Confidence:** 3

**Summary:**

This paper proposes OMOG, an innovative cross-domain graph learning framework designed to enhance the adaptability and performance of Graph Neural Networks (GNNs) across various domains. By training a distinct expert model for each pre-training graph and employing adaptive gating functions during inference, OMOG dynamically selects relevant experts for unseen graphs.

**Strengths:**

1.	By pretraining individual models for each graph dataset, OMOG effectively addresses the feature and structural heterogeneity found across diverse graphs.
2.	OMOG’s model bank allows the easy addition of new expert models without retraining the entire system, providing flexibility to expand the pretraining bank with new data and adapt quickly to novel domains.

**Weaknesses:**

1.	The primary motivation for adopting the “one model for one graph” approach is to alleviate the negative transfer limitations observed in the “one model for all graphs” method. It would be beneficial to provide comparisons and discussions on how this method differs from prior approaches that aim to reduce negative transfer through better pretraining data selection as [1].
2.	It is recommended to identify which models, pretrained on specific graphs, are selected as the best match for various test graphs, with explanations for these selections. Additionally, I’m curious about whether pretraining data from different domains can contribute effectively or if only similar/same-domain data is more beneficial would strengthen the analysis. A case study is recommended to evaluate whether the proposed gating strategy actually mitigate issues stemming from conflicts across pre-training data from diverse domains.
3.	It would be valuable to explore whether this pipeline could be adapted to other self-supervised learning approaches for pretraining graph-specific experts, and additional ablation studies are expected.
4.	OMOG’s design requires a separate model for each dataset and can result in a large model bank when many datasets are involved. Will it lead to high storage costs and maintenance overhead, especially in resource-constrained environments? A complexity analysis would also be helpful to understand OMOG’s computational feasibility at scale.

[1] Better with less: a data-active perspective on pre-training graph neural networks. In NIPS '23.

**Questions:**

see Weaknesses

---

> ### Author Response · Authors · 2024-11-20
>
> >**Q1.** The primary motivation for adopting the “one model for one graph” approach is to alleviate the negative transfer limitations observed in the “one model for all graphs” method. It would be beneficial to provide comparisons and discussions on how this method differs from prior approaches that aim to reduce negative transfer through better pretraining data selection as [1].
>
> **A1.** Thank you for your question. The data selection approach such as [1] implements data selection in the pre-training stage. After the pre-training, the model is fixed and applied to all the downstream tasks. This method will exclude knowledge from the pretraining data and would cause generalization difficulty on downstream datasets. Compared to this approach, our method is more flexible. We do not eliminate any knowledge from the pretraining data but store all in the bank. During inference, we adaptively choose the most relevant models (knowledge) for a downstream dataset (or task), which encourages the positive knowledge transfer while mitigating the negative transfer more effectively.
>
>
> >**Q2.** It is recommended to identify which models, pretrained on specific graphs, are selected as the best match for various test graphs, with explanations for these selections. Additionally, I’m curious about whether pretraining data from different domains can contribute effectively or if only similar/same-domain data is more beneficial and would strengthen the analysis. A case study is recommended to evaluate whether the proposed gating strategy actually mitigates issues stemming from conflicts across pre-training data from diverse domains.
>
> **A2.** Thank you for your question. We have provided a case study in Section 4.5.  We consider the 9 downstream datasets for zero-shot node classification. Specifically, we select 10 random samples from each dataset and visualize the average relevance score given by different gate functions. From the figure in Section 4.5, we find that in most of the situations, the gates will choose expert models trained in similar domains, which indicate that datsets from similar domain will contribute more in general. But in several cases, the gate will choose models from different domains. For example, Sports dataset of e-commerce domain have the highest relevance scores to the Wikics dataset in academic datasets, which indicates a sign of cross-domain transfer in certain situation.
>
> >**Q3.** It would be valuable to explore whether this pipeline could be adapted to other self-supervised learning approaches for pretraining graph-specific experts. and additional ablation studies are expected.
>
> **A3.** Thank you for your suggestion. In the table below we attach the results attained by switching self-supervised approaches to GraphMAE. As a comparison, we find that both the original GRACE method and GraphMAE methods could attain similar performance. We will add the results in the revision.
>
> | Methods | Child | History | Cora | Citeseer | Dblp | Products | Pubmed | Sports | Wikics |
> |---| ---| --- | ---| ---| ---| --- | ---| ---| ---|
> |w/ GraphMAE|20.78| 26.51| 64.28| 49.03| 56.88| 31.97| 39.89| 23.65| 61.54|
> |w/ GRACE|20.34| 25.68| 66.19| 49.23| 57.53| 31.02| 39.71| 23.54| 62.42|
>
>
> >**Q4.** OMOG’s design requires a separate model for each dataset and can result in a large model bank when many datasets are involved. Will it lead to high storage costs and maintenance overhead, especially in resource-constrained environments? A complexity analysis would also be helpful to understand OMOG’s computational feasibility at scale.
>
> **A4.** Thank you for your question. Our models are light-weighted. The storage cost of one pair of expert and gate models is 579KB. Considering the storage capability of current devices, the cost is relatively low and it is feasible to maintain the model bank.
>
> For a single expert model, the training time complexity is $O(d^2+K^2d)$, where $K$ is the number of hops aggregated for a target node and $d$ is the dimension of the feature vector. For a single gate model, the training time complexity is also $O(d^2+K^2d)$. Thus, the pretraining time complexity on a single dataset is just $O(d^2+K^2d)$. This manifests the feasibility of scaling up our frameworks. Moreover, since our framework allows us to train multiple experts and gates on different datasets parallely, it actually would gain efficiency advantages compared to the previous cross-domain pretraining methods.

---

> > ### Comment · Reviewer_6z4L · 2024-11-28
> >
> > Thank you for your further explanations and for providing more details. While some of my concerns have been addressed, I still have the question about the motivation for selecting models. The authors argue that choosing models can prevent the drawbacks associated with data selection, such as the exclusion of some knowledge and the difficulties in generalization. However, since different expert models are trained on different datasets independently and only K of them are chosen, could this approach also lead to the elimination of knowledge?

---

> > > ### Author Response · Authors · 2024-11-29
> > >
> > > > Thank you for your further explanations and for providing more details. While some of my concerns have been addressed, I still have the question about the motivation for selecting models. The authors argue that choosing models can prevent the drawbacks associated with data selection, such as the exclusion of some knowledge and the difficulties in generalization. However, since different expert models are trained on different datasets independently and only K of them are chosen, could this approach also lead to the elimination of knowledge?
> > >
> > > Thank you for your comment. We choose top-K relevant models to boost the positive knowledge transfer, while the less relevant models tend to contain irrelevant knowledge which possibly causes negative transfer. This is evidenced by our experiments in Figure 5, in which selecting most irrelevant models would lead to significant performance drops. Moreover, we show the model performance in Figure 6 with varying the value of K. It demonstrates that when the number of models K exceeds 3, the model performance does not improve and may even decline.This indicates that, within our framework, model selection is unlikely to eliminate valuable knowledge but instead prevents negative transfer.

---

### Official Review · Reviewer_Fbbv · 2024-11-03

**Soundness:** 3
**Presentation:** 3
**Contribution:** 3
**Rating:** 5
**Confidence:** 4

**Summary:**

This paper proposes a novel cross-domain pretraining framework called "one model for one graph," by pretraining a bank of expert models and using a gating function to choose a subset of experts to effectively integrate prior model knowledge.

**Strengths:**

1. The presentation of this paper is good and most parts of the paper are clear.
2. This paper proposes a novel cross-domain pretraining framework.
3. The experimental results demonstrate the effectiveness of the proposed method.

**Weaknesses:**

1. The intuition of the generator and filter in Pretraining the gate module is not clear.
2. The authors lack the discussion about the difference between the proposed method and the mixture-of-experts based methods.
3. I am concerned about the negative transfer issue in the proposed method. Since the knowledge in the proposed methods is extracted from graphs in different domains (and most of them are irrelevant), it inevitably increases the probability of facing the negative transfer issue. How does the proposed method address this issue? The top-k strategy seems to only filter out the low confidant knowledge, while it can not directly alleviate the negative transfer issue as the irrelevant knowledge might be included in the top k expert models.
4. I try to reproduce the experimental results, but there is no instruction and datasets available in the provide GitHub link.

**Questions:**

1. What is the time complexity or running time of the proposed method given that the proposed method needs to pretrained the model on several graphs from different domains?
2. In line 207, the authors mention that the node-level feature are randomly maked, resulting in two maked views. Are these two masked views mutually exclusive? For instance, given a 10-d feature matrix, the first masked view is generated by masked 5 features and the second view is generated by masking the rest 5 features?
3. How do you ensure that the generator only generates a matrix to mask the domain irrelevant features such that the filter features are domain-related?
4. The authors mention that one key issue is negative transfer. Since the knowledge in the proposed methods is extracted from graphs in different domains, it inevitably increases the probability of facing the negative transfer issue. How does the proposed method address this issue? The top-k strategy seems to only filter out the low confidant knowledge, while it can not directly alleviate the negative transfer issue as the irrelevant knowledge might be included in the top k expert models.
5. I try to reproduce the experimental results, but there is no instruction and datasets available in the provide GitHub link. The readme seems to be empty. Could you provide the datasets and the instruction to reproduce the results?

---

> ### Author Response · Authors · 2024-11-20
>
> > **Q1.** What is the time complexity or running time of the proposed method given that the proposed method needs to pretrained the model on several graphs from different domains?
>
>
> **A1.** Thank you for your question. For a single expert model, the training time complexity is $O(d^2+K^2d)$, where $K$ is the number of hops aggregated for a target node and $d$ is the dimension of the feature vector. For a single gate model, the training time complexity is also $O(d^2+K^2d)$. Thus, the pretraining time complexity on a single dataset is just $O(d^2+K^2d)$. Moreover, since our framework allows us to train multiple experts and gates on different datasets in parallel, it actually would gain efficiency advantages compared to the previous cross-domain pretraining methods.
>
>
> > **Q2.** In line 207, the authors mention that the node-level feature are randomly maked, resulting in two maked views. Are these two masked views mutually exclusive? For instance, given a 10-d feature matrix, the first masked view is generated by masked 5 features and the second view is generated by masking the rest 5 features?
>
> **A2.** Thank you for your question. For the contrastive learning on node features, we follow the approach of GRACE to mask the features randomly. We would pinpoint this detail in the revision.
>
>
>
> > **Q3.** How do you ensure that the generator only generates a matrix to mask the domain irrelevant features such that the filter features are domain-related?
>
> **A3.** Thank you for your question. For the generated matrix $a_i$, we tend to let it learn to mask the domain-irrelevant features through the two loss terms $dis(\tilde{f},f_{center})$ and $\frac{1}{dis(\mathbf{o_i,f_{center}})}$ in Equation (4), where $\tilde{f}$ is embedding of original features masked by $a_i$ in the output space, $o_i$ is embedding of $a_i$ in the output space, and $f_{center}$ can be viewed as the representative feature of the training domain. With the loss terms decreasing, the distance between the mask matrix $o_i$ and $f_{center}$ will become larger but $\tilde{f}$ masked by $o_i$ will become closer to the $f_{center}$. Hence, $a_i$ are supposed to learn to mask the domain-irrelevant features of the input vector.
>
>
>
> > **Q4.** The authors mention that one key issue is negative transfer. Since the knowledge in the proposed methods is extracted from graphs in different domains, it inevitably increases the probability of facing the negative transfer issue. How does the proposed method address this issue? The top-k strategy seems to only filter out the low confidant knowledge, while it can not directly alleviate the negative transfer issue as the irrelevant knowledge might be included in the top-k expert models.
>
> **A4.** Thank you for your question. We agree that the negative transfer is inevitable in cross-domain graph learning, and our proposed framework aims to mitigate the level of negative transfer. Specifically, we train the gates and let them to select top-k models that are most relevant to the new task. To deal with possible negative transfer within the top-k models, the gates would calculate the relevant scores for the k models, and their parameters will be weighted according to the score. This is validated by our experiments in Figure 6. When increasing the number of experts, the variance of the performance of top-k strategy is minor. This indicates that even including irrelevant knowledge, the performance will not be affected because of the low relavent score used in weighting. In this way, the proposed method tends to let the most relevant model contribute the most and thus alleviate the possible impacts of irrelevant knowledge included in the top-k models.
>
> > **Q5.** I try to reproduce the experimental results, but there is no instruction and datasets available in the provide GitHub link. The readme seems to be empty. Could you provide the datasets and the instruction to reproduce the results?
>
> **A5.** Thank you for your question. The dataset sizes are too large to upload to the anonymous github repo. We are trying to re-organize them into smaller ones and will provide them and the instructions shortly.

---

> > ### Comment · Reviewer_Fbbv · 2024-11-24
> > **Reply to Authors' Rebuttal**
> >
> > Thank you for the detailed explanation. You address most of my concerns. However, I check the paper and it seems that the paper is not updated, such as the confusion about the masking, how to select 10 sample mentioned by Reviewer 6eQR, etc. If you have already done it, please highlight the changes. In addition, when can you provide the instruction and dataset for reproducing the results?

---

> > > ### Author Response · Authors · 2024-11-26
> > >
> > > Thank you for your comment. We are glad that we have addressed your concerns. We have updated the PDF file of the paper and the instructions for both the datasets and the codes.

---

### Official Review · Reviewer_6eQR · 2024-11-04

**Soundness:** 1
**Presentation:** 3
**Contribution:** 2
**Rating:** 5
**Confidence:** 5

**Summary:**

This paper proposes the OMOG (One Model for One Graph) framework, which advances graph learning by pre-training a unique model for each graph within a model bank. By creating a bank of expert models, each pre-trained for a specific dataset, OMOG selects relevant experts for inference using gate modules tailored to the target graph’s domain. This approach mitigates negative transfer issues common in multi-domain pre-training, and it performs effectively in zero-shot and few-shot tasks like link prediction and node classification, showing its potential for cross-domain graph tasks.

**Strengths:**

1. The paper focuses on and attempts to address a crucial yet highly challenging problem in the field of graph analysis—constructing a Graph Foundation Model (GFM).
2. On commonly used graph datasets, the model OMOG presented in the paper achieves relatively good performance in both zero-shot and few-shot settings.

**Weaknesses:**

1. The paper does not clearly explain the differences from other MOE-based methods, such as GraphAlign and AnyGraph. The approach seems very similar to these methods, leaving it unclear what specific advantages OMOG has over them and why it achieves improved performance.
2. A core idea of OMOG is that each pre-training dataset requires a dedicated expert. This approach poses challenges for scalability: as the volume of pre-training data increases, the model grows linearly with the data, which is detrimental to pre-training efficiency.
3. Why is the expert model specifically a Transformer? How would the performance change if other models, such as GNN, Graph Transformer, or MLP, were used instead? Additionally, prior to entering the experts, the features and structure are fused through SGC. Why couldn’t this fusion step be incorporated within the experts themselves? After all, different graphs may require varying levels of neighbor aggregation.
4. The core part for achieving zero-shot in this paper relies on calculating the similarity between label embeddings and prediction embeddings to obtain the final label prediction. In fact, most models that work under few-shot settings can be adapted to zero-shot using a similar approach. Consequently, Table 1 lacks several relevant baselines, such as GraphAlign, GCOPE, and GraphMAE.
5. Do all experts contribute to downstream performance improvements? In Figure 6, while the number of experts is adjusted, the full set of pre-training data is still used to train the gating mechanism. Could you vary the number of pre-training datasets to examine how this affects downstream performance?
6. Although this paper discusses GFM, which should be applicable to various downstream tasks, there is still an absence of experiments on graph-level tasks, such as graph classification or graph regression.
7. Some parts of the paper lack clarity. For example, in Section 4.5, the phrase ‘select 10 samples from each dataset’ is ambiguous. Does this refer to selecting 10 nodes, subgraphs, or something else?

**Questions:**

See weaknesses.

---

> ### Author Response · Authors · 2024-11-20
>
> >**Q1.** The paper does not clearly explain the differences from other MOE-based methods, such as GraphAlign and AnyGraph. The approach seems very similar to these methods, leaving it unclear what specific advantages OMOG has over them and why it achieves improved performance.
>
> **A1.** Thank you for the question. Previous MOE methods have the following characteristics: (1) Have a fixed number of experts with a model. (2) Pretrain all the experts on different datasets together. (3) Apply the one pretrained model for all downstream datasets. Compared to the previous datasets, our new method has following advantages: (1) Train different expert models on different datasets parallelly, avoiding the potential conflicts from different data domains during pretraining. (2) Flexibly choose suitable experts for different downstream tasks, which boosts the positive knowledge transfer while mitigating the negative transfer. Furthermore, as empirical evidence shown in the table below, our proposed method outperforms the two methods on the node classification tasks when the datasets are numerous and from different domains.
>
> | Methods | Child | History | Cora | Citeseer | Dblp | Products | Pubmed | Sports | Wikics |
> |---| ---| --- | ---| ---| ---| --- | ---| ---| ---|
> |Anygraph     |13.84 |15.16| 55.63| 40.03| 50.27| 22.36 |37.92 |15.35| 50.84|
> |GraphAlign|18.63| 26.39| 63.45| 45.11| 55.11| 30.82| 37.43| 21.73| 60.17|
> |OMOG      |20.34| 25.68| 66.19| 49.23| 57.53| 31.02| 39.71| 23.65| 62.42|
>
> >**Q2.** A core idea of OMOG is that each pre-training dataset requires a dedicated expert. This approach poses challenges for scalability: as the volume of pre-training data increases, the model grows linearly with the data, which is detrimental to pre-training efficiency.
>
> **A2.** Thank you for your question. We would like to point out that to construct the pre-trained model bank, it is not required to train a large model on various datasets. Instead, our approach only trains light-weight expert and gate models on each dataset, respectively. Hence, the pre-training process on multiple datasets can run in parallel, which actually provides our method efficiency advantages over other cross-domain pre-training approaches.
>
>
> >**Q3.** Why is the expert model specifically a Transformer? How would the performance change if other models, such as GNN, Graph Transformer, or MLP, were used instead? Additionally, prior to entering the experts, the features and structure are fused through SGC. Why couldn’t this fusion step be incorporated within the experts themselves? After all, different graphs may require varying levels of neighbor aggregation.
>
> **A3.** Thank you for your question. We would like to point out that our method can fuse the different levels of neighborhood. Specifically, we deploy SGC of different layers to aggregate different hops of neighborhood information. Then we apply the self-attention module to adaptively fuse the varying hops of features. Compared to MLP, our backbone applies SGC to capture the graph structure information. Compared to GNN and GPS (graph transformer), our backbone can selectively assign attention weights to aggregated embeddings from different hops, which is more suitable to be generalized to different datasets (different dataset might require a different number of aggregation layers). As a comparison, we also test the performance of zero-shot node classification when using the GCN, GPS and MLP. The results are shown in the table below.
>
> | Methods | Child | History | Cora | Citeseer | Dblp | Products | Pubmed | Sports | Wikics |
> |---| ---| --- | ---| ---| ---| --- | ---| ---| ---|
> |GCN |18.36| 21.65| 60.71| 44.91| 50.28| 30.63| 37.47| 21.83| 59.14|
> |GPS |17.58| 21.35| 57.50| 43.81| 55.05| 28.42| 33.92| 19.27| 58.23|
> |MLP |19.27| 23.27| 65.36| 47.94| 53.68| 29.30| 40.16| 22.98| 62.02|
> |OMOG|20.34| 25.68| 66.19| 49.23| 57.53| 31.02| 39.71| 23.65| 62.42|
>
> From the table, we could find out that our expert backbone outperforms all other models.

---

> > ### Author Response · Authors · 2024-11-20
> >
> > >**Q4.** The core part for achieving zero-shot in this paper relies on calculating the similarity between label embeddings and prediction embeddings to obtain the final label prediction. In fact, most models that work under few-shot settings can be adapted to zero-shot using a similar approach. Consequently, Table 1 lacks several relevant baselines, such as GraphAlign, GCOPE, and GraphMAE.
> >
> > **A4.** Thank you for your question. We add the performance of GraphAlign, GCOPE and GraphMAE when using the cosine similarity as the output layer. We find that in most cases, our model can get better results on both node classification and link prediction than these baselines. We will add the results to Table 1 in the revision.
> >
> > Results for zero-shot node classification:
> > | Methods | Child | History | Cora | Citeseer | Dblp | Products | Pubmed | Sports | Wikics |
> > |---| ---| --- | ---| ---| ---| --- | ---| ---| ---|
> > |GraphMAE  |15.37| 20.63| 62.83| 46.78| 51.27| 26.23| 33.95| 20.72| 57.98|
> > |GCOPE     |16.83| 20.37| 61.48| 44.26| 53.40| 27.75| 34.26| 19.88| 55.23|
> > |GraphAlign|18.63| 26.39| 63.45| 45.11| 55.11| 30.82| 37.43| 21.73| 60.17|
> > |OMOG      |20.34| 25.68| 66.19| 49.23| 57.53| 31.02| 39.71| 23.65| 62.42|
> >
> >  Results for zero-shot link prediction:
> >   | Methods | Child | History | Cora | Citeseer | Dblp | Products | Pubmed | Sports | Wikics |
> >  |---| ---| --- | ---| ---| ---| --- | ---| ---| ---|
> > |GraphMAE  |22.35| 24.15| 51.68| 43.78| 52.28| 33.21| 42.15| 34.20| 49.24|
> > |GCOPE     |24.33| 25.83| 50.24| 44.84| 47.24| 34.85| 45.53| 33.54| 47.29|
> > |GraphAlign|31.81| 32.28| 52.92| 51.34| 51.21| 38.05| 44.28| 35.53| 50.12|
> > |OMOG      |31.29| 34.86| 56.28| 50.72| 53.46| 40.95| 49.42| 37.81| 52.38|
> >
> >
> > > **Q5.** Do all experts contribute to downstream performance improvements? In Figure 6, while the number of experts is adjusted, the full set of pre-training data is still used to train the gating mechanism. Could you vary the number of pre-training datasets to examine how this affects downstream performance?
> >
> > **A5.** Thank you for your question. We would like to clarify that our gates are also dataset-specific, just like the experts. This indicates that we do not use all datasets to train one gate, instead, we also train a bank of gates on different datasets respectively. Thus, in Figure 6, by varying the number of experts, we are actually just varying the number of pretrained datasets used in the downstream tasks. As the results manifest, not all experts contribute to the downstream performance.
> >
> > > **Q6.** Although this paper discusses GFM, which should be applicable to various downstream tasks, there is still an absence of experiments on graph-level tasks, such as graph classification or graph regression.
> >
> > **A6.** Thank you for your question. We would like to clarify that we do not explicitly claim that our proposed method is a graph foundation model in the paper. The main focus of our research is to mitigate the negative transfer of cross-domain learning on text-attributed graphs. Thus, the scope of our method is limited to the node-level and link-level tasks, which usually have text-attributed graph data. Still, we thank the reviewer for the comment and will seek opportunity to extend our framework to graph-level tasks in the revision.
> >
> >
> >
> > > **Q7.** Some parts of the paper lack clarity. For example, in Section 4.5, the phrase ‘select 10 samples from each dataset’ is ambiguous. Does this refer to selecting 10 nodes, subgraphs, or something else?
> >
> > **A7.** Thank you for your questions. By selecting 10 samples, we refer to selecting the ego-subgraphs of 10 randomly sampled nodes. We will clarify this in the revision.

---

> > > ### Comment · Reviewer_6eQR · 2024-11-25
> > >
> > > Thank you for your supplementary explanations and experiments during the rebuttal period. These addressed some of my concerns, and I have raised my score to 5. However, I still have the following three key issues:
> > >
> > > 1. You highlighted as an advantage of your method that “Train different expert models on different datasets parallelly, avoiding the potential conflicts from different data domains during pretraining.” However, I find this reasoning unclear. If conflicts exist among different data domains, the expert models trained on these domains independently should also inherit these conflicts. In such a case, there should be no need to combine multiple experts during downstream tuning.
> > >
> > > 2. You mentioned that pretraining on multiple datasets can be done in parallel. Could you report the training time and GPU memory consumption of your method compared to the baselines?
> > >
> > > 3. In your response to Q3, you stated that “we deploy SGC of different layers to aggregate different hops of neighborhood information.” However, Figure 2 and the description on line 197 of your paper suggest that you repeatedly apply Equation 2, which does not imply using different numbers of repetitions of Equation 2 for different datasets. In fact, on some datasets, increasing the number of repetitions does not necessarily improve performance. Additionally, the experiment you provided in your response is not what I intended to see. My suggestion was to replace the Expert model with other architectures, not to replace the entire pre-training model with a different one.
> > >
> > > If these issues can be clarified through further discussion, I will consider raising my score further.

---

> ### Author Response · Authors · 2024-11-26
>
> > 1. You highlighted as an advantage of your method that “Train different expert models on different datasets parallelly, avoiding the potential conflicts from different data domains during pretraining.” However, I find this reasoning unclear. If conflicts exist among different data domains, the expert models trained on these domains independently should also inherit these conflicts. In such a case, there should be no need to combine multiple experts during downstream tuning.
>
> Thank you for your comment. Based on the previous research [1][2], graph datasets from similar domains are more likely to produce positive transfer, while datasets from different domains tend to produce negative transfer. Hence, if all datasets are pretrained together, the model would inevitably suffer negative transfer. Instead, our framework will first pretrain the expert models on different datasets, which prevents the negative transfer during pretraining. Moreover, for inference on a new dataset, the framework will **adaptively choose experts of high relevance** to merge, thus mitigating the conflicts/negative transfer to a large extent and boosting the positive transfer. This is addressed by our experiments in Figure 5, when merging all the models or random k models, the performance will drop due to the conflict/negative transfer compared to merging the top-k models. Hence, combining top-k relevent experts is useful for downstream task infering.
>
> Reference: [1] Text-space Graph Foundation Models: Comprehensive Benchmarks and New Insights, Chen et al.
>
> [2] Better with Less: A Data-Active Perspective on Pre-Training Graph Neural Networks, Xu et al.
>
> > 2. You mentioned that pretraining on multiple datasets can be done in parallel. Could you report the training time and GPU memory consumption of your method compared to the baselines?
>
> Thank you for the comment. We list training time and GPU memory consumption in the following table, which manifests that our method has advantages in efficiency due to the lightweight models and parallel framework.
>
> | | GraphMAE | Oneforall | LLaGA | AnyGraph | GraphAlign | ZeroG | OMOG |
> |---| ---| --- | ---| ---| ---| --- | ---|
> |Time            |3.7h| 12.3h| 20.3h| 9.8h| 3.8h| 21.6h|  0.9h|
> |GPU Memory      |8.3G| 17G| 35.6G| 14.5G| 6.7G| 39.6G|  4.1G|
>
> > 3. In your response to Q3, you stated that “we deploy SGC of different layers to aggregate different hops of neighborhood information.” However, Figure 2 and the description on line 197 of your paper suggest that you repeatedly apply Equation 2, which does not imply using different numbers of repetitions of Equation 2 for different datasets. In fact, on some datasets, increasing the number of repetitions does not necessarily improve performance. Additionally, the experiment you provided in your response is not what I intended to see. My suggestion was to replace the Expert model with other architectures, not to replace the entire pre-training model with a different one.
>
> Thank you for your comment. There are some misunderstandings we would like to clarify. For the SGC aggregations, we keep the ego-subgraph embeddings $h^{(\alpha)}$ after each repetition. This will result in a list of embeddings containing different hops of information, $[h^{(1)}, h^{(2)}, ..., h^{(\alpha)}]$, of a target node as stated in line 205. We did not just repeat aggregations and only kept the final embedding. Then we apply the self-attention module to adaptively fuse these embeddings of different hops since different datasets might favor different levels of aggregations. In this way, our expert model is flexible and can be applied to different datasets.
>
> Regarding the experiments in Answer 3, what we did is indeed replacing the expert model backbone with other architecture, not replacing the whole model. We are sorry that we did not specifically enhance this point in the answer and caused the confusion. If the reviewer has more questions, we are more the glad to answer.

---

> > ### Comment · Reviewer_6eQR · 2024-11-26
> >
> > Thank you for your detailed response and clarification.
> >
> > I noticed that both in [1] and [2], comparisons are provided against “using semi-supervised method training from scratch (e.g. gcn, gin, etc)”. However, I am curious why Table 2 in your paper does not include a direct comparison with methods that is trained from scratch on downstream datasets.
> > As widely known, basic semi-supervised learning achieves good performance on these datasets (e.g., results on Cora can exceed 80%), while your method only achieves 75.41 with extensive pre-training. Additionally, in terms of memory usage and computational time, training directly on the downstream graph data from scratch is likely the most cost-effective choice. Could you clarify whether your approach is only better when the downstream dataset does not have any labels?
> >
> > Moreover, in your response to the third part, you mentioned that your method uses a “self-attention module to adaptively fuse these embeddings.” However, I could not find any equations describing this module in the paper. The lack of such crucial details significantly hinders understanding of your method. Even based on your explanation during the rebuttal, if this self-attention module learns different parameters for different datasets, then it should be considered part of the expert model rather than a shared module across all experts. Could you clarify which parts of the model belong to the expert modules and which parts are shared among all experts?
> >
> > I suggest revising the paper to connect the steps of the model more clearly through equations. The current model description in the paper is quite confusing and does not align with the explanations provided in your rebuttal.

---

> ### Author Response · Authors · 2024-11-28
>
> > I noticed that both in [1] and [2], comparisons are provided against “using semi-supervised method training from scratch (e.g. gcn, gin, etc)”. However, I am curious why Table 2 in your paper does not include a direct comparison with methods that is trained from scratch on downstream datasets. As widely known, basic semi-supervised learning achieves good performance on these datasets (e.g., results on Cora can exceed 80%), while your method only achieves 75.41 with extensive pre-training. Additionally, in terms of memory usage and computational time, training directly on the downstream graph data from scratch is likely the most cost-effective choice. Could you clarify whether your approach is only better when the downstream dataset does not have any labels?
>
>
> Thank you for your comment. We would like to point out that [2] does not provide comparisons against semi-supervised methods. All their models and baselines are trained under the unsupervised learning setting (e.g., pretraining + fintuning). Regarding [1], it actually discusses two different settings: co-training (supervised learning) and pertaining (unsupervised learning setting). However, the results from different settings are never compared. Unsupervised learning requires the model to access neither any node label nor the Laplacian matrix of the whole test graph during training, which differs from semi-supervised learning. There have been plenty of works that try to build unsupervised graph learning frameworks. Our work follows this research line and focuses on the unsupervised setting. Building a framework that has superior performance in (semi-)supervised setting is beyond the scope of this paper.
>
>
> > Moreover, in your response to the third part, you mentioned that your method uses a “self-attention module to adaptively fuse these embeddings.” However, I could not find any equations describing this module in the paper. The lack of such crucial details significantly hinders understanding of your method. Even based on your explanation during the rebuttal, if this self-attention module learns different parameters for different datasets, then it should be considered part of the expert model rather than a shared module across all experts. Could you clarify which parts of the model belong to the expert modules and which parts are shared among all experts?
> I suggest revising the paper to connect the steps of the model more clearly through equations. The current model description in the paper is quite confusing and does not align with the explanations provided in your rebuttal.
>
> Thank you for your comment. We indicate the backbone of the expert model is a transformer block (self-attention module) in line 210 and provide the equations in Appendix C. Nonetheless, we agree with the reviewer that the current presentation could lead to certain misunderstandings and we will revise the sections of model descriptions.

---

### Meta-Review · Area_Chair_p7Y9 · 2024-12-23

**Metareview:**

This paper proposes a mixture-of-experts model for improving generalization on graphs that span different domains. The idea is to have a bank of expert models and then route information to select which graph models to use on a new dataset. In contrast to previous MoE-based models, the key innovation of their approach is to have one expert per pretraining dataset.

The reviewers all agreed that the paper addresses an important and timely topic. However, there were concerns about the scalability of the approach, as they need a new model for each pretraining dataset. Additionally, the reviewers raised concerns about whether the method is indeed able to avoid negative transfer as the paper claims.

**Additional Comments On Reviewer Discussion:**

The authors provided replies to many of the reviewer concerns, but some of the key issues surrounding their claims that the model avoids negative transfer were not addressed fully. In the end, none of the reviewers were willing to champion the paper and all of the reviewers agreed that the paper was below the acceptance threshold.

---

### Decision · Program_Chairs · 2025-01-22

Reject